# Peer review of "Mathematical Framework for Wearable Devices in the Internet of Things Using Deep Learning"

_diagnostics, 2022, doi:10.3390/diagnostics12112750_

Round 1

Reviewer 1 Report

The authors propose a set of mathematical approaches and a deep learning method for wearable devices in medical applications. The proposed solution is interesting, however, the following recommendations should be taken into account.

1. The paper is not structured well. For instance, there is not any section about related works. SOme indicative works are provided below.

2. Both the abstract and introduction do not provide precisely the scope of the paper. What is the goal of the mathematical approaches and the DL method?

3. The contributions are not also clear. They can be enumerated and briefly described in the introductory part.

4. Regarding the various symbols and notations, a table should be provided, explaining them.

5. A new section about the background of the various method used is necessary.

References

[1] Nahavandi, Darius, et al. "Application of artificial intelligence in wearable devices: Opportunities and challenges." Computer Methods and Programs in Biomedicine 213 (2022): 106541.

[2] Radoglou-Grammatikis, Panagiotis, et al. "A self-learning approach for detecting intrusions in healthcare systems." ICC 2021-IEEE International Conference on Communications. IEEE, 2021.
[3] 
Cheng, Yuemeng, et al. "Recent developments in sensors for wearable device applications." Analytical and Bioanalytical Chemistry 413.24 (2021): 6037-6057.
[4] 
Mukhopadhyay, Subhas Chandra, Nagender Kumar Suryadevara, and Anindya Nag. "Wearable sensors for healthcare: Fabrication to application." Sensors 22.14 (2022): 5137.
[5] 
Khajeh-Khalili, Farzad, and Yasaman Khosravi. "A novel wearable wideband antenna for application in wireless medical communication systems with jeans substrate." The Journal of The Textile Institute 112.8 (2021): 1266-1272.

Author Response

Response sheet attached. thank you

Reviewer 2 Report

I leave suggestions for improving your manuscript:

1. In my opinion there's some redundant information in the introduction the topic could be improved. Also, it could be better to shorten the article's title. There're too many unnecessary details.

2. The architecture of an IoT framework and technology classification is not presented with the proposed idea.

3. Medical application is stated in the title, but in the article, none of the medical services and applications are described

4. In the title, "Wearable Devices" is mentioned, but Wearable sensors or devices for any visualizations are not evaluated.

5. Please use the higher resolution for Figures 1-3.

6. Poor references [1] and [2] are both same.

7. None of the equation concepts are cited.

8. How did they obtain the values presented in the tables?

9. The paper is tough to read, giving the impression that the authors are thinking in another

language while writing in English.

10. Below are some of my primary concerns:

·        The research problem is not well specified.

·        The authors should revise their Abstract to summarize the paper's contents.

·        The Introduction should be revised to more situate and give context to the introduced research subject. The "Related Work" sections should be further separated from the "Introduction" and developed.

·        The authors should add a section at the end of the "Introduction" enumerating the paper's contributions

·        The authors should clearly define the following:

o Compare their input compared to the existing methods.

o   The applicability of their proposed method

·        The authors should explain the following:

o   How did they obtain the values they presented in their tables?

o   The significance of their simulation graphs and how did they get them.

o   What are they trying to prove with their simulation results?

·        Even though the authors worked on many papers, they should base their research on more recent articles.

11. Abstract: is started with a generic form; Abstract need to be improved; it should include the objective of the work and techniques. Also, we need to specify the proper applications of the work

The title seems to be long and should create curiosity. (Capitalised every work, as shown in the template)

Too many Sections are made; try to follow the MDPI structure specified in the template

In section 1 (Introduction), the novelty of the proposed contribution is not sufficiently documented. Related work must be examined in comparison with this work. In addition, a paragraph at the end of the sections, which describes the remainder of the paper, would have been helpful.

12. Results and discussion need to Section: Present critical findings concerning the central research question and Present secondary findings (secondary outcomes, subgroup analyses, etc.)

Author Response

Response sheet attached. thank you

Reviewer 3 Report

Authors have claimed to offer potential methods of detecting various illnesses by using mathematical methods , deep learning and wearable IoT devices. 

1. The work is good. However, I feel that the organization of the paper is not very good, and many concepts lack corresponding explanations.

2. The experiment of the paper does not seem to be enough to support the motivation of the paper and solve the problem of the paper, a few concerns must be addressed.

3. In particular, for some key data and theories, the author should give some citations.

4. The author should analyze some more advanced methods in the related work.

5. Reference no. 1 and 2 are same. Please check.

6. The experiment should further prove the advantages and innovation of the method proposed in this paper.

7. Which IoT simulation tool is used and how all scenario results are being compared?

8.  The reference no. 10, can not be accessed :  10. Ogura, Y.; Parsons, W.H.; Kamat, S.S.; Cravatt, HHS Public Access. Physiol. Behav. 2017, 176, 139–148.

Author Response

Response sheet attached. thank you

Reviewer 4 Report

I have the following comments regarding the paper.

1. The authors need to define the motivation clearly.

2. Some recent literature needs to be inserted.

3. The novelty and contribution is not clear. Please add points of contribution and novelty.

4. The proposed scheme needs to be illustrated in more detail.

5. The results need more clarification and illustration.

6. Figures need improvements

7. Thorough proof read is required for omitting typos, spelling, and grammatical mistakes.

Author Response

Response sheet attached. thank you

Round 2

Reviewer 1 Report

The authors addressed most of the comments adequately. Therefore, the paper can be accepted for publication.

Author Response

Response sheet attached. thanks

Reviewer 2 Report

Most of the previous comments are considered in the current version of the manuscript. But still, few comments are not taken for consideration.

1. The higher resolution figures need to be used. The state of the art of presenting the figures is poor. Figures 1, 2, 3, and 4 must be replaced. 

2. Abstract needs to be improved, it should include the objective of the work, techniques. Also, you need to specify the right applications for the work. Proof reading is required, last sentence of the abstract "68 per cent". Scientific writing needs to be improved. 

Author Response

Response sheet attached. thanks

Reviewer 4 Report

The authors did not provided satisfied answers per my comments. Revise and resubmit.

Author Response

Response sheet attached. thanks

Round 3

Reviewer 4 Report

The manuscript has been improved. I accept.